# Biomimetic Silica Particles with Self-Loading BMP-2 Knuckle Epitope Peptide and Its Delivery for Bone Regeneration

**DOI:** 10.3390/pharmaceutics15041061

**Published:** 2023-03-25

**Authors:** Mi-Ran Ki, Thi Khoa My Nguyen, Tae-In Park, Hae-Min Park, Seung Pil Pack

**Affiliations:** 1Department of Biotechnology and Bioinformatics, Korea University, Sejong 30019, Republic of Korea; 2Institute of Industrial Technology, Korea University, Sejong 30019, Republic of Korea; 3Department of Chemical Engineering and Applied Chemistry, Chungnam National University, Daejeon 34134, Republic of Korea

**Keywords:** BMP2, BMP2 knuckle epitope residues, P4 peptide, biomimetic silica deposition, drug delivery, controlled release, bone regeneration

## Abstract

Biomimetic silica deposition is an in-situ immobilization method for bioactive molecules under biocompatible conditions. The osteoinductive P4 peptide derived from the knuckle epitope of bone morphogenetic protein (BMP), which binds to BMP receptor-II (BMPRII), has been newly found to contain silica formation ability. We found that the two lysine residues at the N-terminus of P4 played a vital role in silica deposition. The P4 peptide co-precipitated with silica during P4-mediated silicification, yielding P4/silica hybrid particles (P4@Si) with a high loading efficiency of 87%. P4 was released from P4@Si at a constant rate for over 250 h, representing a zero-order kinetic model. In flow cytometric analysis, P4@Si showed a 1.5-fold increase in the delivery capacity to MC3T3 E1 cells than the free form of P4. Furthermore, P4 was found anchored to hydroxyapatite (HA) through a hexa-glutamate tag, followed by P4-mediated silicification, yielding P4@Si coated HA. This suggested a superior osteoinductive potential compared to silica or P4 alone coated HA in the in vitro study. In conclusion, the co-delivery of the osteoinductive P4 peptide and silica by P4-mediated silica deposition is an efficient method for capturing and delivering its molecules and inducing synergistic osteogenesis.

## 1. Introduction

The FDA-approved bone morphogenetic protein 2 (BMP2) is currently used to treat bone defects because of its excellent osteoinductive activity [1]. However, the high cost of recombinant BMP2 protein and its adverse effects limit its clinical application, except for a few orthopedic disease treatments [2].

The tertiary structures of proteins are important in terms of drug design and discovery. Based on the tertiary structure, molecular modeling helps identify the critical binding site required for protein–protein interactions, enabling the identification and design of local sites involved in protein–ligand interactions using the structure of the entire protein [3,4]. Therefore, it is possible to develop peptide drugs that can be readily manufactured via chemical synthesis rather than using the entire protein [5,6].

The P4 peptide comprises a modified sequence, KIPKASSVPTELSAISTLYL, corresponding to residues 73–92 of the knuckle epitope of bone morphogenetic protein 2 (BMP2), which is part of the BMP-binding site of BMP receptor II (BMPRII) [7,8,9]. The knuckle epitope of BMP 2 binds to the extracellular domain of BMPR II, and then the intracellular domain of BMPR II phosphorylates the type I receptor, which induces intracellular SMAD signaling [10]. BMP/SMAD signaling plays a critical role in the formation and maintenance of bone, cartilage and other tissues [11]. This P4 peptide has already been demonstrated to enhance bone formation, and it is believed to be due to the mechanisms above [7,8,9,12].

However, similar to proteins, peptides also have low bioavailability and short half-lives owing to their low stability during proteolysis and aggregation in aqueous solutions, limiting their practical use [6,13,14]. To overcome these challenges, peptide delivery systems, chemical modifications of peptide structures and combinations of the two have been investigated. If the bonding between the carrier and peptide is based on physical adsorption, delivery efficacy and targeting remain to be resolved. Many studies have focused on increasing the immobilization efficiency of the delivery system and the induction of the controlled release of peptides by adopting affinity tags or covalent cross-linkage [15,16,17,18,19,20,21,22]. For example, seven aspartic acids or glutamic acids have been introduced to the peptide to increase binding to hydroxyapatite (HA), a bone component [15,18,20,21,23], and the induction of controlled release under redox conditions through disulfide linkage by introducing thiol residues has been attempted [16,17]. Mussel-inspired adhesion molecules, such as dopamine, have been introduced into carriers to increase their affinity for peptides [24,25]. To combat proteolysis, the cyclization of peptides by introducing cysteine residues was evaluated [26]. These approaches can help reduce the number of peptides applied, owing to their improved binding affinity, and provide sustained peptide release. However, there may also be a risk of inactivation or an immune response due to peptide modification [27].

Silica particles have been studied as carrier systems for drug delivery because of their biocompatibility, stability, porous structure and ease of surface functionalization [28,29]. Particularly, biomimetic silica condensation, wherein peptides or proteins capable of silica deposition produce silica nanoparticles from silicic acid under ambient conditions, has received significant attention as a system that could efficiently load and deliver payloads in-situ. [30,31,32,33,34,35]. Previously, we demonstrated that BMP2 could deposit biomimetic silica [36]. BMP2 co-precipitated with silica improved its encapsulation efficiency in the graft material compared to its physical adsorption onto the graft and effectively delivered itself to cells, showing an increase in osteogenic gene expression and calcium mineralization. The heparin-binding domain (or motif) (HBM) at the N-terminus of BMP2 plays a role in its silica precipitation activity of BMP2 [36]. However, the silica formation activity of the BMP2 protein was 17 times higher than that of HBM with the same number of molecules [36]. This suggests that the protein structure itself or other parts of the protein also contributes to the silica-forming ability of BMP2.

As described above, the knuckle epitope (P4) of BMP2 has a potential osteoinductive activity. The improved stability and efficient delivery of the silica-self-entrapped BMP2 protein to cells prompted us to investigate whether the P4 peptide has a silica-forming ability. In this study, we identified and characterized the silica-forming ability of P4. Based on these properties, the drug delivery pattern, efficiency and osteoinductive activity of self-entrapped P4 in silica particles were evaluated in vitro.

## 2. Materials and Methods

### 2.1. Materials

The recombinant human BMP2 (rhBMP2) was supplied by CG-Bio (CG Bio Inc., Seoul, Republic of Korea). The peptides named R5 [30], Ectp1 [37], KPS [38], KPT [39], P4 [7], P4 variant peptide and HBM [36] were synthesized by Peptron (Peptron, Dajeon, Republic of Korea), and Fluorescein isothiocyanate (FITC)-labeled P4 (FITC was conjugated to the C-termini of peptide) and P4E × 6 were synthesized by GenScript (Piscataway, NJ, USA). The synthetic peptides used in this study were obtained from a supplier, having undergone purification via HPLC and subsequent LC/Mass analysis. The molecular weight measured by mass spectrometry was consistent with the theoretical value. The purity of the peptide was 95%. The resulting mass data along with peptide sequences for each peptide are listed in Table 1. The tetramethyl orthosilicate (TMOS), L-ascorbic acid, β-glycerophosphate, ammonium heptamolybdate, oxalic acid, Alizarin Red-S and hydroxyapatite (HA) powder (<200 nm particle size) were purchased from Sigma-Aldrich (St. Louis, MO, USA). Collagen sponges were purchased from Olympus Terumo Biomaterials Co. (Tokyo, Japan). CellTracker Red CMTPX (Invitrogen) was purchased from Thermo Fisher Scientific Korea Ltd. (Seoul, Republic of Korea). Mounting medium with 4′,6-diamidino-2-phenylindole (DAPI) (abcam) was purchased from Dawinbio Inc. (Hanam, Republic of Korea) All the other reagents were of analytical grade.

### 2.2. Silica Deposition

The silica deposition catalyzed by peptide (100 μg) was initiated by adding 10 μL of 1 M TMOS in 1 mM HCl to 90 μL of 100 mM sodium phosphate (pH 7.6) buffer containing peptide or protein, based on the method in Luckarift et al. [40]. Alternatively, the reaction was performed in 50 mM tris(hydroxymethyl)aminomethane (Tris)-HCl (pH 8.0) with or without 100 mM NaCl to characterize the silica deposition ability of P4. The reaction was allowed to proceed at room temperature for 1 h. In the control assay, peptides or phosphate ions were absent in the reaction mixture. The silica precipitate was washed thrice using ddH_2_O by centrifugation (10,000× *g*, 10 min, 4 °C).

### 2.3. Silica Quantification

As described previously, the amount of silica precipitated by the peptides was determined using a modified molybdenum blue method [36]. Briefly, the washed silica precipitate was dried, dissolved in 1 M NaOH solution and diluted 5- to 10-fold for a total reaction solution of 50 μL. To this mixture, an equal volume of 1% ammonium heptamolybdate, 1% oxalic acid and 100 mM of ascorbic acid were added sequentially. Oxalic acid prevents the formation of molybdophosphoric acid while generating β-silicomolybdate complexes [41], and ascorbic acid is used to enhance color development. The absorbance of the blue solution was measured at 810 nm.

### 2.4. Co-Precipitation of P4 with Silica, Yielding P4-Encapsylated Silica Particle (P4@Si) by P4-Catalyzed Silicification and Measurement of Entrapment and Loading Efficiency

A total of 100 μg of P4 peptide containing 10% FITC-labeled P4 (P4-FITC) was used to precipitate silica particles as described above (Figure 1A). The amount of co-precipitated peptide was calculated by subtracting the amount of P4 remaining in the supernatant after removing the precipitates from the initial feeding amount. To compare immobilization and release of P4 by silica co-precipitation, 100 μL of P4 peptide (1 mgmL^−1^) was bound to 1 mg of collagen sponge or 1 mg of HA powder for 1 h, which were suspended in 1 mL of PBS, centrifuged and isolated from the supernatant. The total released and loaded amounts of P4 were measured at 485/535 nm using a fluorescence microplate reader (Infinite F200 NanoQuant; TECAN, Männedorf, Switzerland) and corrected by considering P4-FITC as 10% of the P4 released. The amount of silica was determined using the modified molybdenum blue method described above. The entrapment efficiency (%) and loading efficiency (%) of P4 were calculated, respectively, based on the equations below:$$\mathrm{Entrapment}\;\mathrm{Efficiency}\;\left(\mathrm{EE}\right)\left(\%\right)=\frac{mass\;of\;bound\;P4}{initial\;mass\;of\;P4\;added}\times 100$$
$$\mathrm{Loading}\;\mathrm{Efficiency}\;\left(\mathrm{LE}\right)\;\%=\frac{mass\;of\;bound\;P4}{total\;mass\;of\;delivery\;system}\times 100$$

### 2.5. Determination of Release Kinetics

The three kinds of P4 bound carriers were incubated for varying durations in 0.5 mL of PBS at 37 °C with shaking at 150 rpm. The P4 released from the silica particles, collagen sponge or HA powder was collected over 250 h. After removing the supernatant, the delivery system was re-dispersed in fresh PBS (0.5 mL of new PBS for the next release and the fluorescence intensity of the collected P4 containing P4-FITC was monitored at 485/535 nm using a fluorescence microplate reader. To analyze the P4 release pattern from carriers, the cumulative amount of P4 released at each time point was fitted using the following equation based on the Korsmeyer–Peppas equation [42]:$$\log_{10}(\frac{M_{t}}{M_{\infty }}\times 100)=\log_{10}k+n\log_{10}t$$
where *M_t_* is the cumulative amount of P4 released at time *t*; *M*_∞_ is the cumulative amount of P4 released at infinite time; *k* is the release rate constant; and *n* is the diffusional exponent [43].

### 2.6. Determination of Silica Release from P4@Si

The amount of released silica recovered from P4@Si at the indicated time was quantified according to the silica quantification method after drying 100 μL of the supernatant before dissolving in 1 M NaOH.

### 2.7. MC3T3 E1 Cell Culture

Mouse preosteoblast MC3T3 E1 subclone 4 (ATCC^®^ CRL-2593™, Manassas, VA, USA) was obtained, and passage 16 cells were cultured in a growth medium (GM) containing Minimal Essential Medium alpha (MEM-α, without L-ascorbic acid, WELGENE, Gyeongsan-si, Gyeongsangbuk, Republic of Korea), supplemented with 10% fetal bovine serum and 1% Penicillin-Streptomycin solution (Invitrogen) at 37 °C in a humidified atmosphere with 5% CO_2_. For osteogenic differentiation, GM was replaced with bone differentiation medium (BM) [44,45], which contains GM along with 50 µg mL^−1^ L-ascorbic acid and 10 mM β-glycerophosphate but lacks dexamethasone. Since dexamethasone has been known to promote the differentiation of mesenchymal stem cells into osteoblasts [46], it was removed from the differentiation medium to investigate the osteogenic potential of the BMP2 peptide. The BM was replaced every three days.

### 2.8. Flow Cytometry Analysis of Cellular Uptake of Free P4 and P4@Si

Each P4 preparation (P4-FITC accounts for 10% of the total 50 µgmL^−1^ of P4) was treated on MC3T3 E1 in Opti-MEM medium (GIBCO, without phenol red) for 4 h and subsequently washed thrice in PBS after cell harvesting by trypsinization. Cells were resuspended in cold 70% ethanol and kept on ice for analysis or −20 °C for subsequent analysis. Flow cytometry was performed using AttuneTM NxT Flow Cytometer, blue (Invitrogen). FITC was used to confirm the cellular uptake of free P4 and P4@Si using FITC conjugated P4. The forward- and side-scatter voltages and FITC fluorescence were optimized until the main population of the dot signal reached approximately 2 × 10^2^. The flow cytometry data were gated to eliminate noise signals, such as cell debris and non-specific molecules. The flow rate for measurement was 25 µLmin^−1^, and the measurement was stopped when 10,000 events of the gated cell were captured.

### 2.9. STED Microscopy and Confocal Images of MC3T3 E1 Cells

MC3T3 E1 cells on a coverslip in a 12-well plate supplemented with Opti-MEM were incubated with P4-FITC or P4-FITC@Si in a humidified atmosphere containing 5% CO_2_ at 37 °C for 4 h. The cells were washed 3 times with PBS and fixed with 4% paraformaldehyde at 37 °C for 30 min. After washing the cells three times with PBS, the cells were immersed in PBS containing CellTracker Red CMTPX (Thermo Fisher Scientific Korea Ltd. (Seoul, Republic of Korea) (5000:1 dilution) for 30 min to stain the cytoplasm. The stained cells on the cover slip were washed 3 times with PBS and applied on the mounting drop with DAPI on a slide glass. The confocal images were taken along the XY plane and *z*-axis using a STEDYCON microscope (abberior instruments GmbH, Göttingen, Germany) with a 100× oil immersion objective lens. MC3T3 E1 cells show rhodamine-labeled cytoplasm (red), FITC-labeled p4 peptide or P4-silica particles (green), and DAPI-labeled nucleic acids (cyan hot). Laser power was set at 10%, and the pixel size was 30 nm. Z-stacks of the cells were acquired with a 100 nm step size. ImageJ software was used to generate videos of the z-stacks and 3D reconstructions of the confocal images.

### 2.10. Preparation of HA Nanocomposite Powders (Figure 1B)

The sterile HA powder (10 mg) (<200 nm particle size) was mixed with 0.1 mL of polyglutamate (E6) linked to the C-terminus of P4 peptide, namely P4E × 6 peptide solution (0.2 mgmL^−1^) and then allowed to react for 10 min. Unbound peptides were eliminated by gentle rinsing. Next, P4E × 6-bound HA or HA powder was reacted in 1 mL of PBS containing 20 mM hydrolyzed TMOS overnight with gentle agitation. The P4E × 6-bound HA and HA were prepared without additives. All preparations were harvested by centrifugation and washed thrice using ethanol (first) and distilled water. The resulting product was resuspended in 1 mL of sterile PBS.

### 2.11. Cell Proliferation

A total of 100 μL of MC3T3 E1 osteoblastic cells in GM (1 × 10^5^ cellsmL^−1^) were seeded onto a well of 96-well plates and then cultured for 24 h; the medium was changed with BM in the absence or presence of HA composite. The BM was changed every 3 days and cultured for one week. The viability of the cells cultures of each well was measured using the 3-(4,5-dimethylthiazol-2-yl)-5-(3-carboxymethoxyphenyl)-2-(4-sulfophenyl)-2H-tetrazolium (MTS) assay (the CellTiter 96^®^ Aqueous One Solution Cell Proliferation assay kit (Promega)), according to the manufacturer’s instructions. The absorbance of the mixture was recorded at 490 nm using a UV/visible microplate reader (Infinite M200 PRO NanoQuant; TECAN).

### 2.12. Osteogenic Gene Expression

Quantitative PCR (qPCR) was performed as previously described [30] to evaluate the expression of osteogenic differentiation markers, including alkaline phosphatase (ALP) (forward: ACCGCTGCCCGAATCC; reverse: TCTCCTCGCCCGTGTTGT) and osteocalcin (OCN) (forward: GTGAGCTTAACCCTGCTTGTGA; reverse: TGCGTTTGTAGGCGGTCTTC) in MC-3T3 E1 cells on Day 7 and on Day 14 after induction of bone differentiation using the BM in the presence of HA composites. The amount of target genes was expressed as relative quantification (RQ) over the expression level of control HA on Day 7. The quantitative gene expression data were normalized to the expression levels of Glyceraldehyde-3-phosphate dehydrogenase (GAPDH) (forward: CCTGGCCAAGGTCATCCATG; reverse: GCAGGAGACAACCTGGTCCT) [47].

### 2.13. Calcium Deposition and Quantification

MC3T3 E1 cells were cultured in GM for 24 h under the same conditions as mentioned above. The medium was changed to BM, and calcium deposition was measured after 21 days. The BM was replaced every three days. Calcium deposition was determined using an Alizarin Red S assay. The cells were washed with PBS without Ca^2+^ and Mg^2+^ twice, fixed in 4% neutral buffered paraformaldehyde for 30 min, rinsed with ddH_2_O three times to eliminate phosphate ions and stained using 2% Alizarin Red S aqueous solution (pH 4.2 for 30 min). The stained cells were rinsed with ddH_2_O thrice and dried. The cells were observed under a phase-contrast microscope. The area of the calcium-precipitated site was measured using ImageJ software (National Institutes of Health, Bethesda, MD, USA).

### 2.14. SEM and EDX

The structure and composition of P4@Si were analyzed using variable pressure field emission scanning electron microscopy (VP-FE-SEM) and energy-dispersive X-ray spectrometry (EDX), respectively (KBSI, Chuncheon, Republic of Korea).

### 2.15. Statistical Analysis

All values are presented as the mean ± SD. Statistical analyses were performed using Student’s *t*-test to determine the differences between the two groups. A one-way analysis of variance (ANOVA) (https://goodcalculator.com/one-way-anova-calculator/) (accessed on 20 December 2022) was used for multiple comparisons. Statistical significance was set at *p* < 0.05.

## 3. Results

### 3.1. Silica Deposition Ability of Osteoinductive BMP-2 Knuckle Epitope Peptide (P4)

The synthetic peptide P4 corresponding to the knuckle epitope residues 73–92 of the BMP2 protein, its sequence and other silica-forming peptide sequences are summarized in Table 1.

Each peptide-mediated biomimetic silica was prepared from TMOS-derived silicic acid in sodium phosphate buffer. A comparison of the amount of silica produced using each method is shown in Figure 1A. P4 showed a silica deposition ability of approximately 50% compared to the R5 peptide, which is well-known as a silica-forming peptide [30,48,49], and KPS, KPT and the heparin-binding motif of BMP2 (HBM), which was confirmed in our previous studies [36,39] (Figure 1A). The SEM image displayed raspberry-shaped silica particles with sizes of 100–300 nm, which were formed by the agglutination of small silica particles (Figure 1B). EDX analysis showed the existence of C and N, which are organic components of the peptide, indicating that the formed silica particles are a hybrid of the peptide and silica.

### 3.2. Characterization of P4-Mediated Silica Deposition

The P4 peptide catalyzed the silica condensation reaction of silicic acid in sodium phosphate and Tris-HCl buffers. Although silica precipitation was not observed when only NaCl was present without the buffer, silica precipitation increased when NaCl was present in the buffer solution (Figure 2A). Although NaCl is not an absolute requirement for P4-induced silica precipitation, NaCl in Tris-HCl increased silica precipitation by approximately two times that in the absence of NaCl. To identify which residues in the P4 sequence play an essential role in silica deposition, P4 variants with modified residues were prepared, and their silica condensation abilities were compared (Figure 2B). KP4A, in which lysine residues are substituted with alanine, exhibits up to 90% lower silica deposition activity than P4. When the C-terminal LYL sequence was removed, precipitation activity was reduced by 60%. The SP4G peptide, in which serine or threonine with OH sidechain was substituted with glycine, showed an approximately 20% decrease in silica precipitation activity.

### 3.3. Peptide Release from Self-Entrapped Silica Particles

The time-dependent release pattern of the P4 peptide from co-precipitates of P4 and silica (P4@Si) was investigated using a P4 peptide containing 10% P4-FITC. Furthermore, the release rates after P4 adsorption onto collagen sponge (CLG) or HA powder, which is commonly used as a BMP2 carrier in the simple immersion method, were compared. The entrapment efficiency (EE%) and loading efficiency (LE%) for each carrier were calculated as shown in Section 2.4 and Table 2.

The entrapment efficiency (EE%) and loading efficiency (LE%) for each carrier were calculated as shown in Section 2.4 and Table 2. The EE% of the P4 peptide was 77.30% and 51.58% for collagen (P4/CLG) and HA (P4/HA), respectively. Self-entrapped P4 in silica exhibited an EE of 68%. To load 100 μg of P4, 1 mg of either collagen or HA was used. The loading efficiencies of P4 were 7.15% for collagen and 4.91% for collagen and HA, respectively. In contrast, the P4 in the self-assembled silica hybrid exhibited a LE of 87.18%. The cumulative release of P4 peptide from each loaded material is shown in Figure 3A. P4 adsorbed onto collagen (P4/CLG) was released 21.5% after 1 h, 80% after 60 h; the remaining quantity was released slowly. P4 adsorbed on HA (P4/HA) was released at 51% after 1 h and 87% over 60 h and was hardly released. Although P4 in HA was released faster than P4 in collagen within 1 h, the latter was released faster (Figure 3A).

Self-captured P4 in silica showed a low release rate of 2.8% after 1 h and a cumulative release of approximately 23% after 60 h. Based on the Korsmeyer–Peppas model equation (Table 3), the rate constant (*k*) was the highest for P4/HA and the lowest for P4@Si and showed an intermediate value for P4/CLG. The diffusional exponent (*n*) of P4@Si was greater than 0.5, indicating non-Fickian diffusion. The other values were <0.5, indicating a Fickian diffusion. The cumulative silica release rate of P4@Si was similar to that of P4 (Figure 3B), which was calculated from the amount of P4 released amount from P4@si over time were $(\sum_{t=0}^{t}P4\%)=0.2193t+6.1001,\;R^{2}=0.9772$, and the silica release rate was $f(\sum_{t=0}^{t}Si\%)=0.2206t+0.2315,\;R^{2}=0.9299$.

### 3.4. Enhancement of Cellular Uptake of P4 through Self-Entrapped Silica Carrier

The cellular uptake of free P4 and P4@Si was confirmed by flow cytometry (Figure 4A). The number of treated cells that had P4-FITC uptake was significantly improved by self-entrapped P4 in silica, yielding 1.5-fold cells exhibiting P4 uptake compared to free P4 (3873 ± 56 for free P4 vs. 5473 ± 141 for P4@Si, *p* < 0.0001) (Figure 4B). The confocal images revealed that P4-FITC@Si particles bind more to cells than P4-FITC (Figure 4C,D). Free form of P4-FITC was located not only inside of cells and the cell surface as an aggregated form (white arrow in Figure 4C). Additionally, z-axis imaging displayed that the silica particles bind to the cell surface (Figure 4D, lower image).

### 3.5. Synergistic Effect of the Combination of P4 with Silica on Osteogenesis

In this study, to ensure that silica particles were evenly spread onto the carrier HA surface, a hexa-glutamate tag was conjugated to the C-terminus of P4 to locate it on the HA surface first, followed by a P4-mediated silica formation reaction on the HA surface, which generates P4@Si coated HA. A silica coating with silicic acid in the absence of organic matter was prepared as a control. To determine the biocompatibility of the HA preparations, the viability of MC3T3 E1 cells was measured 7 days after treatment with each preparation (Figure 5A).

Compared to bare HA, the cell viability increased by more than 40% and 15% in silica-coated HA (Si/HA) and P4E × 6@Si/HA, respectively. However, there was no significant difference between HA with attached P4E × 6 peptide only (P4E × 6/HA) and bare HA. Cells treated with silica-coated HA showed an increased expression of both ALP and OCN on Day 7 (Figure 5B,C). There was an increase in ALP expression 1.5-fold and 3-fold in Si/HA and P4E × 6@Si/HA, respectively, whereas OCN expression increased 6-fold and 28-fold, respectively. On Day 14, ALP expression decreased in the P4E × 6@Si/HA group compared to that in the control group. OCN sharply increased in the silica-containing group on Day 7 and continued to show increased expression on Day 14, increasing to more than twice as high in P4E × 6@Si/HA compared to the others (Figure 5C). HA with the P4 peptide simply adsorbed showed an approximately 1.4-fold increase in ALP with no significant differences, and OCN expression with significant differences on Day 14 compared to control HA. Increased calcium precipitation was observed on silica-coated HA rather than on P4/HA, with the highest calcium mineralization observed on P4 with silica (P4E × 6@Si/HA) (Figure 5D,E). Meanwhile, calcium minerals in the control HA appeared to have smaller HA grains gathered; however, those in P4E × 6/HA showed larger and denser precipitates (Figure 5E).

## 4. Discussion

BMP2 is a crucial protein for inducing bone formation. It achieves this by differentiating mesenchymal stem cells into osteoblasts/osteocytes, which then produce bone tissue [50]. The P4 peptide, consisting of residues 73–92 of the BMP2 knuckle epitope, induces ALP activity, but a concentration 500 times higher than that of BMP2 protein is required for comparable activity (Appendix A). In particular, freshly prepared P4 peptides could induce ALP enzyme activity, and there was no concentration-dependent increase in activity between the two concentrations (1 and 10 µM), in part due to aggregation or instability in aqueous solutions [6,13,14]. Based on the silica self-immobilization method for stabilizing BMP2 protein [36], the study explored whether this technique could be applied to the BMP2 peptide.

We characterized the silica-forming ability of the osteoinductive-P4 peptide. P4@Si formed by the co-precipitation of the peptide and silica showed a structure similar to the silica particles formed by silica-forming peptides and proteins in previous studies [36,39]. The P4 peptide catalyzed the silica condensation reaction of silicic acid in sodium phosphate and Tris-HCl buffers. NaCl presence in Tris-HCl increased silica precipitation by approximately twice as that without NaCl. Although silica gel was formed at low ionic strengths, silica particles were successfully prepared at high ionic strengths in the presence of NaCl [51]. The presence of salt likely reduced the repulsive forces between molecules, allowing them to assemble into silica particles [52]. Tris bases have been reported to catalyze the synthesis of colloidal silica and to serve as carbon sources for mesoporous carbon fabrication [53]. In another study, the amine group of Tris was used for the surface amination of silica particles by reacting with a silanol group [54]. This suggests that phosphate ions, NaCl and Tris bases can act as accelerators to promote P4-mediated silica condensation. The P4 variant, KP4A, in which lysine residues were substituted with alanine, exhibited as much as 90% lower silica deposition activity than P4, indicating that the lysine residue plays an essential role in P4-mediated silicification. The positively charged ε-amino groups of the lysine residues in P4 peptides and negatively charged phosphate ions are supposed to mediate the formation of siloxane bonds resulting in precipitated silica as proved in R5 peptide [48,49]. In the R5 peptide, the C-terminal sequence of RRIL is important for the self-assembly of the peptide and is known to play a role in creating nuclei for silica deposition to increase particle size [48]. The LYL portion of P4 is thought to play a role in this self-assembly. Serine residues play an essential role in silica precipitation through electrostatic interactions with polyamines and are phosphorylated in natural silacidins in diatoms [55]. The SP4G peptide, in which serine and threonine with OH sidechain were substituted with glycine, showed an approximately 20% decrease in silica precipitation activity. Even in synthetic peptides without the phosphorylation of the OH group by post-translational modification, the hydroxyl group plays a vital role in hydrogen bonding with the silica precursor, which also affects silica formation [56]. In terms of silicification, it is confirmed that serine and threonine make a lower contribution to P4-mediated silicification than lysine residues or C-terminal LYL sequences. Various supramolecular assemblies can be constructed by hydrogen bonding between building blocks, as shown in proteins and DNA. Therefore, the hydrogen bond between P4 and silica is expected to at least partially play a vital role in the growth of this organic-inorganic composite.

Because of its ability to form silica, P4 was self-entrapped in silica with high efficiency (LE of 87.18%), showing a sustained release pattern from the silica. P4 adsorbed onto collagen (P4/CLG) or HA (P4/HA) showed a burst release at an early incubation time within 60 h, caused by collagen polymer expansion and degradation [57] or a lack of attraction between P4 and HA (Figure 3A and Table 3). Compared with simple adsorption, the initial burst release was suppressed in P4@Si. The release was observed at a constant rate for 250 h, indicating a zero-order kinetic model (R^2^ = 0.9772) (Figure 3A). Based on the Korsmeyer–Peppas model (Table 3), the rate constant (*k*) related to the initial release rate was 10–20 times lower for P4-immobilized silica than for P4 attached by physical adsorption. In addition, the diffusional exponent (*n*) of P4@Si was greater than 0.5, indicating that a continuous release was possible owing to diffusion and erosion, even when the residual P4 was at a low level. Given that the slopes of cumulative release over time were similar for P4 and silica, the release of P4 was sustained by diffusion and silica erosion (Figure 3B).

The study found that P4 peptide delivered through silica particles was more effective in targeting cells than free P4. This was evidenced by the fact that a larger proportion of P4 was delivered to the target cells through P4-entrapped silica particles compared to the free form of P4 (Figure 4A,B). Under STED microscopy, the use of P4@Si nanoparticles allowed for better targeting and delivery of P4 to cell as it was visible on the surface of cell (Figure 4D), whereas the free form of P4 was observed in an aggregated form outside the cell (white arrow in Figure 4C) and some inside the cell. Peptides with a positive hydrophobicity scale tend to aggregate in an aqueous solution, and so does the P4 peptide [13,58]. Silica nanoparticles, which are biocompatible and hydrophilic, have been shown to be a promising drug delivery system due to their ability to penetrate tissue systems, facilitate drug uptake by cells and ensure efficient drug delivery to the target site [31,32,33,35,59,60]. Because of their hydrophilic properties, silica nanoparticles can block opsonization and easily move through the blood system [61]. Hence, silica nanoparticles are suitable for capturing and delivering peptides that aggregate in aqueous solutions.

Combining BMP2 with an osteoconductive carrier can induce bone regeneration in the defect area to the same extent as an autologous bone graft [62]. A previous study confirmed that the BMP2 protein was immobilized on the HA surface with a higher yield than simple adsorption through BMP2 protein-mediated silica deposition [36]. Similarly, the P4 peptide was immobilized on HA surface through P4-mediated silica deposition. However, the amount of bound peptide to HA was minimal, resulting in limited silica deposition on the surface, which was unevenly localized. To overcome this issue, a two-stage approach was adopted, which involved immobilizing the peptide onto the HA surface in the first stage and subsequently inducing silica deposition on it. For this purpose, modified P4 (P4E × 6) was synthesized with a linker sequence comprising glycine and serine between the P4 peptide and the polyglutamate residues (E × 6). A glutamate tag at the C-terminus of P4 binds to HA, allowing the peptides to have directionality and exposing the silica-forming N-terminus to the outside to effectively form silica [63,64]. As shown in Appendix A, the quantity of peptide bound to the HA surface was 1.5-fold higher in P4E × 6 than in P4, and the quantity of silica synthesized on the surface by the bound peptide also increased accordingly. P4E × 6 caused a five-fold increase in silica precipitation compared to P4 peptide. This is thought because the hexa-glutamate tag, which binds to HA instead of lysine residues, makes free lysine residues available for silica formation, thus increasing the likelihood of their participation in silica deposition. The FT-IR spectrum analysis of each HA composite confirmed the bonding of P4E × 6 and the formation of silica [65,66]. The intense absorption peak observed at 1080 cm^−1^ in the p4E × 6 peptide was notably diminished in HA composites, suggesting that the binding of the peptide to calcium ions of HA may be responsible for this observation [67]. In the case of P4E × 6@Si/HA, specific peaks for amide I at 1651 cm^−1^ and -OH vibrations at around 3200–3500 cm^−1^ for P4 peptide and silica, respectively, were observed, while no specific peaks for each coating component were detected in the cases of the peptide-only and silica-only coatings (Appendix A).

HA doped with P4-immobilized silica particles drastically increased osteogenesis compared with HA coated with P4 or silica alone. On Day 7, Si/HA showed a 1.5-fold increase, and P4E × 6@Si/P4 exhibited over a 2.5-fold increase in ALP expression compared to the control, while P4E × 6/HA showed a 0.5-fold decrease. ALP is an enzyme present at high levels in the early stages of osteoblast differentiation where cells actively synthesize and deposit extracellular matrix components including collagen and other proteins essential for bone mineralization [68]. ALP expression and activity can be stimulated by either BMP2 or (bio)silica in osteoblasts [69,70]. Biosilica may enhance ALP expression through increasing BMP2 expression in MC3T3 E1 cells. In Western blotting (Appendix A), P4/pSMAD signaling was activated depending on each HA composite. Phospho-SMAD protein level increased in biosilica, 1 µg of P4E × 6, and both 1 and 5 µg of P4E × 6@Si after 4 h of exposure, and in all P4E × 6 presence groups after 8 h. The results indicate that the use of the appropriate amount of peptide is essential as a higher amount of P4E × 6 reduced the signal. Biosilica increased the level of pSMAD even in the absence of P4E × 6, which is thought to be related to the synergistic effect of the P4 combined with biosilica. The results also suggest that silica immobilization can reduce toxicity by preventing direct exposure of high-concentration peptides. The P4 peptide induces SMAD signals [71], and according to literatures, the Kd value of P4 for BMPR type II was different, depending on the experimental conditions used: 18.3 µM in a fluorescence spectroscopy assay [9] and 8.16 × 10^−2^ M using surface plasmon resonance (SPR) technology [26]. On the other hand, the Kd value of BMP2 protein for BMPR II has been reported to be about 50~100 nM using SPR technology [72]. The difference in affinity between the P4 and BMP2 proteins for BMPRII explains the difference in their osteogenic activities.

As shown in Figure 5, on Day 7, the silica-coated HA showed significantly increased cell proliferation, and ALP and osteocalcin (OCN) expressions compared to P4E × 6/HA and the control group. This was consistent with the calcium mineralization on Day 21. The lower ALP level in P4E × 6/HA on Day 7 may be due to delayed proliferation or suppressed pSMAD signaling (Appendix A) compared to the control group. However, by Day 14, P4E × 6/HA showed a 1.4-fold increase in ALP expression compared to the control group owing to P4, which can induce osteogenic pSMAD signaling pathway. In contrast, ALP expression was highest on Day 7 and then decreased on Day 14 in P4E × 6@Si/HA, suggesting that the osteoblasts treated with P4E × 6@Si/HA had matured into osteocytes. As the osteoblasts continue to differentiate and the extracellular matrix becomes more mineralized, the cells undergo a transformation into osteocytes with reduced metabolic activity, which can lead to a decrease in ALP expression as shown in Figure 5B [73]. Meanwhile, OCN is produced by mature osteoblasts and osteocytes [74,75]. OCN significantly increased in the silica-existing groups on Day 7 and continued to show increased expression on Day 14, especially more than twice as high in P4E × 6@Si/HA compared to the others (Figure 5C). In parallel, increased calcium precipitation was observed on HA with silica, either alone or in combination with P4, rather than with P4 alone, with the highest calcium mineralization on HA with P4-immobilized silica (P4E × 6@Si/HA) (Figure 5D,E). The biological activity of HA has been enhanced by substituting Si ions with HA lattice phosphate sites [76]. Kim et al. showed that, compared to non-substituted HA, Si-substituted HA promotes human stem cell attachment and increases cell proliferation, DNA content and osteogenesis-related genes (ALP, Runx2 and OCN) [77]. When the strong osteoinductive BMP2 protein was delivered as a silica co-precipitate, a synergistic effect between the two materials was not observed because the activity of BMP2 exceeded that of silica [36,78]. However, in this study, given that the osteoinductive potential of the P4 peptide was 500-fold (Appendix A) or 1000-fold lower than that of BMP2 [13], the osteoinductivity of silica was superior to that of the P4 peptide. Meanwhile, based on a slight increase in ALP and OCN expression and pSMAD signaling in P4E × 6/HA compared to those in Si/HA and control HA in Figure 5 and Appendix A, it is suggested that P4E × 6 contributed to the strong osteogenic activity of P4E × 6 @Si/HA. The osteoinductive potential of P4 alone seemed insignificant, but that of P4-mediated silica was drastically increased compared to either silica or P4 alone in this study. Thus, the P4-mediated self-silica encapsulation method for delivering P4 peptides with low biological activities is effective for peptide drug utilization. Given that the silica substitution of HA has a beneficial effect on bone regeneration, a more detailed study on the synergistic effect is needed when ingredients such as the P4 peptide, which has low activity when used alone, are used together with silica.

## 5. Conclusions

The BMP 2 knuckle epitope P4 peptide was recently discovered as a silica-forming peptide. The P4 peptide self-assembled into silica by co-precipitation with silica during P4-mediated silicification, yielding P4/silica hybrid particles (P4@Si) with an EE of 68% and LE of 87%. P4 trapped in silica was released from the silica particles at a constant rate proportional to the silica erosion rate over 250 h, representing a zero-order kinetic model. It is released without a burst release in the early phase nor stops release in the late phase, owing to the low residual concentration. P4@Si was delivered to MC3T3 E1 cells 1.5 times more than P4 peptide alone. The P4@Si loaded HA yielded superior osteoinductive potential compared to silica or P4 alone coated HA, as well as untreated HA in the in vitro study. Taken together, owing to (1) the combination of osteoinductive silica and P4 and (2) their sustained release, it is thought that the HA surface coated with P4 and silica synergistically promotes bone formation.

## Data Availability

The data presented in this study are available upon request.

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
