# Peer review of "Biomimetic Silica Particles with Self-Loading BMP-2 Knuckle Epitope Peptide and Its Delivery for Bone Regeneration"

_pharmaceutics, 2023, doi:10.3390/pharmaceutics15041061_

Round 1

Reviewer 1 Report

The research study presented in the paper explores the use of biomimetic silica deposition as a technique for immobilizing bioactive molecules, in particular, the P4 peptide derived from the knuckle epitope of bone morphogenetic protein (BMP). The study also discovered that P4 is released from the hybrid particles at a constant rate over 250 hours, which follows a zero-order kinetic model. The research concludes that the co-delivery of the osteoinductive P4 peptide and silica by P4-mediated silica deposition is an efficient method for capturing and delivering molecules, as well as inducing synergistic osteogenesis, suggesting that P4-mediated silica deposition could be a valuable tool for drug delivery and controlled release in bone regeneration. The manuscript is well prepared. I have some minor concerns below that should be addressed before publishing.

1.       Line 168, osteogenic differentiation medium always contains three ingredients: dexamethasone, ascorbic acid, and sodium β-glycerophosphate. In this study, BM contains L-ascorbic acid and β-glycerophosphate. Are there any references?

2.       Line 311 Table 4 should be Table 3.

3.       There are 3 flow cytometry graphs of free P4-FITC and P4-FITC in figure4A, respectively. Do these three graphs present three repeat experiments? If so, please clarify it in the manuscript.

4.       In figure 5B, ALP mRNA expression decreased in P4EX6 group on day 7 comparing with Si and P4E×6@Si/HA. And ALP also decreased in P4E×6@Si/HA on day 14. What is the rationale? More discussions are suggested.

5.       In figure5E, there are two scale bars, a red one and white one. Please remove one.  

6.       Small spelling errors are found in places, for example line 212, Day7 and on Day14 should have a space between day and the numbers. Please double-check the whole manuscript.

Reviewer 2 Report

1-Is it possible to compare the effect of P4 peptide with the positive control (BMP2). Please provide more information on this matter. Additionally, it would be important to study these experiments with different concentrations of peptides on cells.

2- please provide more information about the size characterization of silica peptide.

3- please provide more information in vivo study.

Reviewer 3 Report

Manuscript Number: pharmaceuticals- 2207864

Biomimetic silica particles with self-loading BMP-2 knuckle 2 epitope peptide and its delivery for bone regeneration

The paper by Ki and coworkers discussed the effect of osteoinductive P4 peptide and silica, derived from the BMP2 protein sequence on drug delivery and bone regeneration using a mouse preosteoblast cell line (MC3T3 E1). This is an exciting paper in that the efficient fabrication methods for producing such silica particles remain scarce.

What I am more curious about is why FITC-labeled cells are very low in the cellular uptake analysis (about 1.61-1.92%) in Figure 4A. What happens if the action is prolonged for more than 24 hours?

In addition, the author also used different peptides in different experiments, so why not use the same peptide in all experiments?

The author mentioned in the abstract that the P4 peptide will bind to BMP receptor-II (BMPRII), but in the introduction of this article, it is noted that the P4 peptide forms a complex with BMP receptor 1A. Is P4 bound to BMPRII or BMPR1A?

Can the author further use surface plasmon resonance (SPR) analysis to see the binding ability of the P4 peptide and BMP receptor?

Round 2

Reviewer 2 Report

Thank you , it is better to improve the quality figure4 western blot.